# DISCRIMINATIVE VARIATIONAL AUTOENCODER FOR CONTINUAL LEARNING WITH GENERATIVE REPLAY

## ABSTRACT

Generative replay (GR) is a method to alleviate catastrophic forgetting in continual learning (CL) by generating previous task data and learning them together with the data from new tasks. In this paper, we propose discriminative variational autoencoder (DiVA) to address the GR-based CL problem. DiVA has class-wise discriminative latent embeddings by maximizing the mutual information between classes and latent variables of VAE. Thus, DiVA is directly applicable to classification and class-conditional generation, which are efficient and effective properties in the GR-based CL scenario. Furthermore, we use a novel trick based on domain translation to cover natural images which are challenging to GR-based methods. As a result, DiVA achieved competitive or higher accuracy compared to state-of-the-art algorithms in Permuted MNIST, Split MNIST, and Split CIFAR10 settings.

## 1 INTRODUCTION

Even though deep learning models such as Szegedy et al. (2015); He et al. (2016); Xie et al. (2017) have been showing remarkable performance on many image recognition tasks exceeding human accuracy, they suffer from the catastrophic forgetting problem. The catastrophic forgetting problem occurs when we want to learn several tasks sequentially without any continual learning (CL) technique (Ratcliff (1990); French (1999); Goodfellow et al. (2013)). Theoretically, the catastrophic forgetting problem can be completely avoided if we store all of the past data, but it is sometimes difficult to store all of the previous data due to the limitation on memory capacity.

To handle the forgetting problem in deep learning, some studies suggested weight regularization (WR) techniques such as EWC (Kirkpatrick et al. (2017)), SI (Zenke et al. (2017)), and IMM (Lee et al. (2017)). In these approaches, each of them has a regularization term to constrain the update of some parameters that are important for previous tasks. Other researches such as GEM (Lopez-Paz et al. (2017)), VCL (Nguyen et al. (2018)) introduced an additional memory called coreset which consists of the small number of real images of past tasks to further improve WR algorithm. We will call this approaches as experience replay (ER)-based methods in this paper. Specifically, ER-based methods are useful when we can access previous task data though the amount of past data is limited. Recently, generative replay (GR)-based approaches such as DGR (Shin et al. (2017)) and VGR (Farquhar & Gal (2018)) are also proposed to replay previous tasks with a generative model instead of storing the small number of images into coreset. They showed that if the generative model successfully approximates data distribution, this approach can be a strong way to alleviate the catastrophic forgetting problem.

In this paper, we propose a new efficient and effective model called discriminative variational autoencoder (DiVA) (Fig. 1) to address the GR-based CL problem. GR-based approaches can address a continual learning setting where a stream of real samples is seen only once, which can not be considered in ER-based CL algorithms. However, GR-based approaches have two shortcomings. First, they need an additional module, a generative model, in addition to a classifier. Second, distribution discrepancy between real and generated samples makes GR-based methods hard to address complex natural images such as CIFAR (Krizhevsky & Hinton (2009)) or ImageNet (Deng et al. (2009)). In this point of views, we try to handle the shortcomings. Firstly, distributions of latent variables of DiVA are clustered class-wise discriminatively on the latent space by introducing mutual information maximization between latent variable $z$ and class condition $c$. Thus, DiVA is directly applicable to

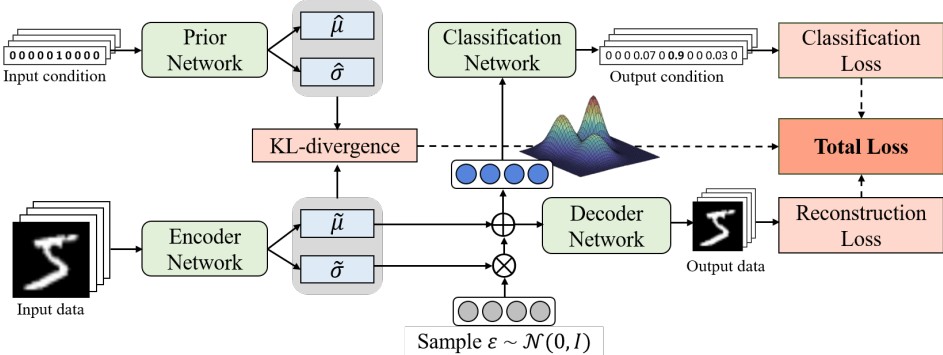

Figure 1: Overview of the discriminative variational autoencoder. The prior network and classification network can be implemented with a simple neural network, e.g., one fully connected layer. The $\oplus$ and $\otimes$ represent element-wise addition and multiplication, respectively.

both classification and class conditional sample generation with one integrated model. Also, we apply a simple but effective domain translation (DT) trick to mitigate the distribution discrepancy and thus DiVA can achieve better classification accuracy on past tasks. To the best of our knowledge, we firstly introduce the DT trick to GR-based CL problem. In summary, we describe three main contributions of our research as follows:

- We suggest a new type of conditional VAE in terms of mutual information maximization, which can be expressed as integrated classifier and learnable prior network. It makes discriminative class-wise distributions for each class on latent space.

- We propose a simple but effective DT trick, which significantly improve CL performance of GR-based approaches on a natural image dataset such as CIFAR10.

- By combining DiVA and DT, we achieved state-of-the-art accuracy on the class incremental learning settings.

## 2 RELATED WORKS

**VAE with GMM prior** VAE (Kingma & Welling (2013)) has been widely studied due to its concrete theoretical background and interesting latent representations. Similar to our mixture-like prior distribution, some researchers proposed learnable Gaussian mixture prior (Dilokthanakul et al. (2016); Nalisnick et al. (2016)) or hierarchical prior distributions (Tomczak & Welling (2018)) because the unit Gaussian assumption as prior in VAE is a too strong constraint. They mainly address an unsupervised learning task while we consider explicit supervision tasks. Wang et al. (2017) also proposed GMM VAE with class-specific and randomly fixed priors for a supervised task. In this case, the random initialization can map similar patterns to far regions on latent space, which could not be suitable for the main desiderata, smoothness, of representation learning (Bengio et al. (2013)). On the other hand, since we consider learnable priors, DiVA satisfies the smoothness condition (Fig. 2(d)).

**Conditional VAE** There are several ways to implement conditional VAE (CVAE). The easiest and intuitive way is to concatenate one-hot vectored class conditions onto latent variables which are outputs of an encoder network (Doersch (2016)). This method sets prior distributions for latent variables of all classes as a unit Gaussian. Since the CVAE always needs data-label pairs for concatenation, other researches propose to infer the class label with an independent heavy classifier (Kingma et al. (2014); Sohn et al. (2015)). As a result, they can make their conditional generative model work with unlabeled data. However, concatenation makes feature representation unnecessarily sparse and requires additional weight parameters for feeding class conditions. In contrast, we do not need the heavy classifier and redundant weights as we model class-wise discriminative latent representations.

**Generative replay** Recently, studies which use GR have been actively proposed to mitigate catastrophic forgetting in the field of CL. This approach generates previous task data when a new task

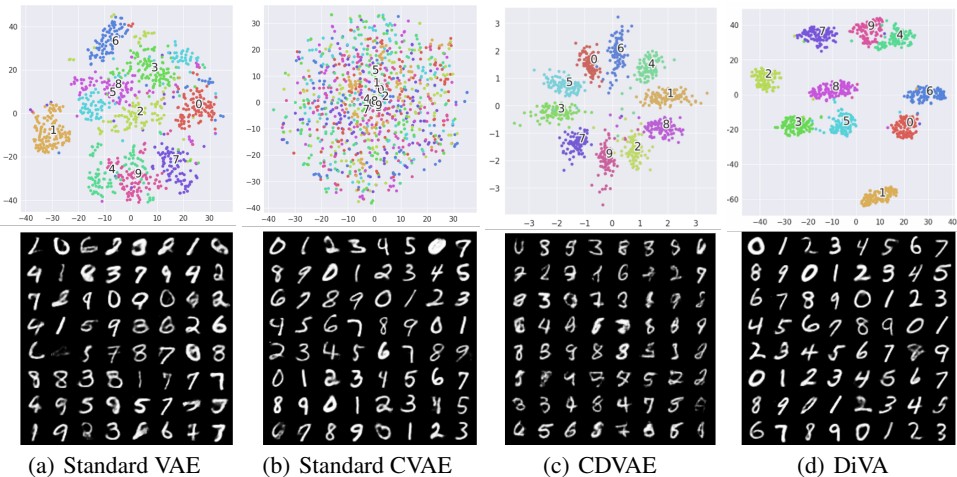

(a) Standard VAE     (b) Standard CVAE     (c) CDVAE     (d) DiVA

Figure 2: Visualization of the latent space (top) and generated samples (bottom). In this visualization, we randomly sampled 1000 images from MNIST test set. We used the t-SNE (Maaten & Hinton (2008)) to reduce the dimension of latent vectors for plotting.

data is coming, then optimizes them concurrently. In the DGR and VGR, they use a GAN-based generator to generate previous task data unconditionally. Since DGR and VGR consider a generator and a classifier as separate modules, they alternately optimize them. Also, other researchers propose GR-based algorithms using VAE such as RtF (van de Ven & Tolias (2018)), CDVAE (Mundt et al. (2019)), and DGDMN (Kamra et al. (2017)). RtF and CDVAE are closely related to our work in that they consider classifier integrated VAE and address CL problem. However, they both assume the prior distribution as unit Gaussian, which contradicts discriminative loss. Also, recently, Hu et al. (2019) (PGMA) proposes a parameter generation and model adaptation trick to mitigate the catastrophic forgetting problem.

## 3 PRELIMINARY

**Variational Autoencoder**    Considering we have observations $X = \{x_1, x_2, ..., x_N\}$ where $x_i \in \mathbb{R}^d$, the goal of the VAE is to maximize marginal log likelihoods of each observation $\log p(x_1, x_2, ..., x_N) = \sum_{i=1}^{N} \log p(x_i)$ with respect to both variational parameters and generative parameters. However, this includes an intractable term when we use a complex model. Because the intractable term is always positive, we can optimize the variational lower bound instead of the marginal log likelihood as follows:

$$\log p(x_i) = D_{KL}[q_\theta(z|x_i)||p_{\theta'}(z|x_i)] + \mathcal{L}(\theta, \theta'; x_i)$$
$$\geq \mathcal{L}(\theta, \theta'; x_i) = \mathbb{E}_{q_\theta(z|x_i)}[\log p_{\theta'}(x_i|z)] - D_{KL}[q_\theta(z|x_i)||p(z)],$$

where $q_\theta(z|x_i)$ is the variational posterior (the encoder) and $p_{\theta'}(x_i|z)$ is the decoder. In general, both the encoder and the decoder can be modeled with neural networks. The $\mathcal{L}(\theta, \theta')$ is called as the evidence lower bound (ELBO).

**Mutual Information**    In information theory, mutual information of two random variables, X and Y, is a measure of the mutual dependence between the two variables. The mutual information can be decomposed into marginal entropy and conditional entropy as follows:

$$I(X;Y) = H(X) - H(X|Y) = H(Y) - H(Y|X).$$

The meaning of this definition is intuitive: if we observe Y then how much uncertainty of X changes. If X and Y are completely independent, i.e. $H(X|Y) = H(X)$, then the $I(X;Y) = 0$. On the contrary, if X is determined by observed Y, then the $I(X;Y)$ is larger than zero.

## 4 DISCRIMINATIVE VARIATIONAL AUTOENCODER

In Fig 2, unconditional generative models such as standard VAE and CDVAE generate some ambiguous samples. Thus, we should consider a conditional generative model for sampling class-wise explicit images to handle the incremental learning problem using a GR-based approach (further details are discussed at Sec.6.1.). Here, we consider a classifier integrated VAE. If latent variables for input images are located class-wise discriminatively on the latent space, we can conduct both class prediction and conditional generation. Fortunately, we can directly model the relationship between latent variables $z$ and class condition $c$ by maximizing mutual information between them. The optimization of DiVA is as follows:

$$\max_{\theta,\theta',\phi,\phi'} \mathbb{E}_{q_\theta(\mathrm{z|x})}[\log p_{\theta'}(\mathrm{x|z})] - D_{KL}[q_\theta(\mathrm{z|x})||p(\mathrm{z})] + \lambda I(\mathrm{c;z}) \tag{1}$$

This can be equivalently rewritten as:

$$\max_{\theta,\theta',\phi,\phi'} \mathbb{E}_{q_\theta(\mathrm{z|x})}[\log p_{\theta'}(\mathrm{x|z})] - D_{KL}[q_\theta(\mathrm{z|x})||\hat{q}_\phi(\mathrm{z|c})] + \lambda\mathbb{E}_{q_\theta(\mathrm{z|x})}[\log \hat{p}_{\phi'}(\mathrm{c|z})] \tag{2}$$

where, $\lambda$ is a weighting parameter of the mutual information loss and $\mathbb{E}_{q_\theta(\mathrm{z|x})}[\log \hat{p}_{\phi'}(\mathrm{c|z})]$ is a cross-entropy for multivariate Bernoullis or L2 loss for continuous-valued conditions. Also, $\theta, \theta', \phi$, and $\phi'$ mean network parameters of the encoder, the decoder, the prior network, and the classifier, respectively. When the encoder and decoder networks are sufficiently complex, it is enough to implement each the prior and classification network as one fully-connected layer. Detailed derivations are described at Sec.4.1.

### 4.1 MUTUAL INFORMATION MAXIMIZATION

In this section, we describe how maximizing $I(\mathrm{z;c})$ with ELBO makes our DiVA. Vincent et al. (2010); Chen et al. (2016) showed that minimization of reconstruction error is related to maximization of mutual information between two variables. Similarly, we maximize mutual information between class condition $c$ and latent variable $z$ by minimizing a reconstruction loss. Here, we will start from $I(\mathrm{c;z}) = H(\mathrm{z}) - H(\mathrm{z|c})$. Then, we will optimize[1]:

$$\begin{aligned} \max I(\mathrm{c;z}) &= \max H(\mathrm{z}) - H(\mathrm{z|c}) \\ &= \max H(\mathrm{z}) + \mathbb{E}_{q_{(\mathrm{c,z})}}[\log \hat{q}(\mathrm{z|c})]. \end{aligned} \tag{3}$$

Now, we derive the conditional entropy $H(\mathrm{z|c})$ as follows:

$$\begin{aligned} H(\mathrm{z|c}) &= -\sum_{\mathrm{c,z}} q(\mathrm{c,z})\log \hat{q}(\mathrm{z|c}), \\ &= -\sum_{\mathrm{c}} q(\mathrm{c}) \sum_{\mathrm{z}} \hat{q}(\mathrm{z|c})\log \hat{q}(\mathrm{z|c}), \end{aligned} \tag{4}$$

If we introduce $\hat{p}(\mathrm{c|z})$, then Eq. (4) can be rewritten as,

$$\begin{aligned} H(\mathrm{z|c}) &= -\sum_{\mathrm{c}} q(\mathrm{c}) \sum_{\mathrm{z}} \hat{q}(\mathrm{z|c}) \left( \log \hat{p}(\mathrm{c|z}) + \log \frac{\hat{q}(\mathrm{z|c})}{\hat{p}(\mathrm{c|z})} \right) \\ &= -\sum_{\mathrm{c}} q(\mathrm{c}) \left( \mathbb{E}_{\hat{q}(z|c)}[\log \hat{p}(\mathrm{c|z})] + D_{KL}[\hat{q}(\mathrm{z|c}) \,||\, \hat{p}(\mathrm{c|z})] \right) \end{aligned} \tag{5}$$

Since $D_{KL}[\hat{q}(\mathrm{z|c}) \,||\, \hat{p}(\mathrm{c|z})]$ is always positive and $q(\mathrm{c})$ is a constant in our dataset, we can get a lower bound of the mutual information as follows:

$$I(\mathrm{c;z}) \geq H(\mathrm{z}) + \mathbb{E}_{\hat{q}(z|c)}[\log \hat{p}(\mathrm{c|z})] \tag{6}$$

where, $\mathbb{E}_{\hat{q}(\mathrm{z|c})}[\log \hat{p}(\mathrm{c|z})]$ can be viewed as class reconstruction in autoencoding manner; minimizing L2 or cross-entropy. In CDVAE, the authors directly optimize $\mathbb{E}_{q(\mathrm{z|x})}[\log \hat{p}(\mathrm{c|z})]$ to impose class information on z so that a classifier can be integrated with VAE as follows:

$$\max_{\theta,\theta',\phi'} \mathbb{E}_{q_\theta(\mathrm{z|x})}[\log p_{\theta'}(\mathrm{x|z})] - D_{KL}[q_\theta(\mathrm{z|x})||p(\mathrm{z})] + \lambda\mathbb{E}_{q_\theta(\mathrm{z|x})}[\log\hat{p}_{\phi'}(\mathrm{c|z})] \tag{7}$$

---

[1]For continuous variables, we can simply change the $\sum$ to $\int$.

---

**Algorithm 1** Train DiVA with domain translation

---

**procedure** Train(DiVA, CyGAN, dt)
  **Input: T** = $\{T\_i\}_{i=1}^{N}$, where $T\_i = \{X_i.Y_i\}$
  // Learning all tasks
  **for all** i = 0, ..., N **do**: // for all tasks
    // Training DiVA
    **for all** (x, y) in $X_i, Y_i$, until convergence **do**:
      sampling $\hat{y}$ // randInt for past task labels
      $\hat{x} \leftarrow$ sampling from $\text{DiVA}_{prev}(\hat{y})$
      $\hat{x} \leftarrow$ CyGAN:s2r($\hat{x}$) **if** dt == s2r **else** $\hat{x}$
      $minimize\ L_{\text{DiVA}}([x, \hat{x}], [y, \hat{y}])$ using Eq. 2
    **end for**
    Test(DiVA, CyGAN, i, dt)
    Train_CyGAN(CyGAN, DiVA, $X_i$)
    $\text{DiVA}_{prev} \leftarrow$ DiVA
  **end for**
  **return** DiVA, CyGAN
**end procedure**

**procedure** Train_CyGAN(CyGAN, DiVA, $X_i$)
  Domain_A $\leftarrow X_i$ // training set X, real images
  **for** i = 0, ..., until given epoch **do**:
    Domain_B $\leftarrow$ sample from DiVA // $\sim$ Task$_i$
    $optimize$ CyGAN(Domain_A $\leftrightarrow$ Domain_B)
  **end for**
**end procedure**

**procedure** Test(DiVA, CyGAN, i, dt)
  **for** j = 0, ..., i **do**: //until current task
    **for all** x in $X_j$ **do**:
      **if** j $\neq$ i: // for past tasks
        x $\leftarrow$ CyGAN:r2s(x) **if** dt == r2s **else** x
      prediction $\leftarrow$ DiVA(x) // classification
    **end for**
  **end for**
**end procedure**

---

However, since they assume unit Gaussian $p(z)$ as a prior for the variational posterior $q(z|x)$, relation between $\hat{q}(z|c)$ and $q(z|x)$ is not considered. Thus, latent space has low density regions because z becomes discriminative within unit Gaussian, which causes ambiguous sample generation (Fig. 2(c)). In contrast, we explicitly consider the relation between $\hat{q}(z|c)$ and $q(z|x)$ as follows. Firstly, since c corresponds to a class among a finite label set, we can simply assume $\hat{q}(z|c)$ as a class-conditional Gaussian distribution, i.e. $\hat{q} \sim \mathcal{N}(z|\mu_c, \sigma_c^2)$. Then, we can replace the unit Gaussian prior in Eq. (1) with the class-conditional Gaussian $\hat{q}(z|c)$, which corresponds to the second term in Eq (2). Now, since we minimize $D_{KL}[q(z|x)||\hat{q}(z|c)]$ in our loss, we can optimize $\mathbb{E}_{q(z|x)}[\log \hat{p}(c|z)]$ instead of $\mathbb{E}_{\hat{q}(z|c)}[\log \hat{p}(c|z)]$. As a result, latent variable z of our DiVA holds information of not only input x but also class c without low density regions: each class condition has its own one mode compact Gaussian distribution. Furthermore, $H(z)$ in Eq. (6) can be indirectly maximized to hold information of x by maximizing the ELBO in Eq. (2). To summarize, our modeling for the probability of the latent variable given its class has three virtues:

- Class is well-separable in the latent space.
- The distribution of the latent variable given its class is simply modeled.
- The latent variable has enough information to reconstruct the image.

The first two virtues are related to the class condition; classification network and prior network, respectively. The last virtue is achieved by the standard VAE.

## 5 DOMAIN TRANSLATION

In the GR-based CL approach, distributions of generated and real images should nearly close to work well. However, there is always a discrepancy between generated and real images, even with current state-of-the-art generative models. For this reason, it is known that GR-based methods are hard to be expanded to a more complex dataset than MNIST variants such as CIFAR10 or ImageNet (Lesort et al. (2018a)). Here, to narrow the distribution discrepancy, we define the two domain, real domain $\mathcal{X}$ and sample domain $\mathcal{Y}$, and their bidirectional domain translation, $G : \mathcal{X} \mapsto \mathcal{Y}$ and $F : \mathcal{Y} \mapsto \mathcal{X}$. In this paper, we call $G$ as a real to sample (r2s) and $F$ as a sample to real (s2r) translation, respectively. We consider two conditions for this translation mappings. First, we should translate only the style as keeping outline patterns of given images. Second, we should consider an unpaired domain translation between real and generated images because the generated images are sampled randomly. In that sense, we use CycleGAN (Zhu et al. (2017)) which satisfies the conditions. When we consider r2s mode, we apply r2s translation to test set images of past tasks and classify them. While, when we consider s2r mode, we translate generated past task samples on the training phase and train our model to classify them. The purpose of both r2s and s2r translation is to narrow the gap between what our model memorizes and what our model looks currently. Detailed definition of this translation is described in Appendix C, and the training process is explained in Algorithm 1.

Table 1: The averaged accuracy of tasks after training of all the 5 tasks are done. WR, ER, and GR mean weight regularization, experience replay, and generative replay, respectively.

| Approach | WR only | | | WR + ER | | GR | | | | |
|---|---|---|---|---|---|---|---|---|---|---|
| Model | EWC | SI | IMM | VCL | GEM | RtF | OCDVAE | DGR | PGMA | DiVA |
| Permuted MNIST | 94.14 | 93.94 | 96.09 | 96.33 | 96.19 | **97.31** | - | 94.54 | 96.77 | 96.01 |
| Split MNIST | 19.8 | 19.67 | 67.25 | 82.71 | 92.20 | 92.56 | 93.24 | 94.8 | 81.7 | **97.92** |

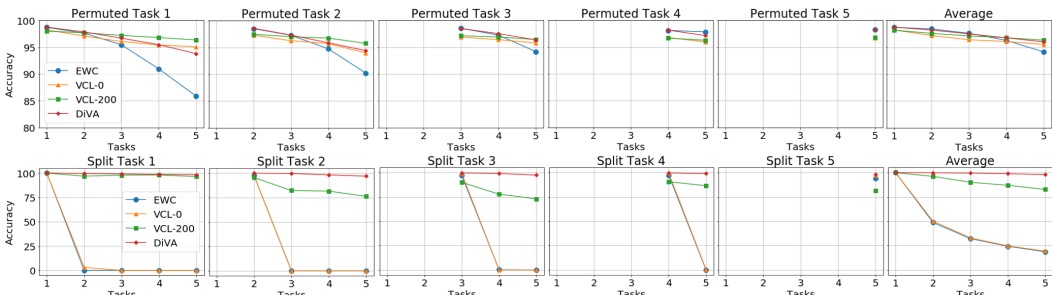

Figure 3: Learning curve for both Permuted (top) and Split MNIST (bottom). Numbers written next to the 'VCL - ' mean the size of random coreset data Nguyen et al. (2018) per each task.

# 6 EXPERIMENTS

In this section, firstly, we discuss the importance of conditional generation for continual learning. Then, we analyze the performance of our model on Permuted MNIST and Split MNIST. Finally, we discuss the effectiveness of our domain translation trick for GR-based methods to address CIFAR10 dataset. Since most of the GR-based approaches did not report results on Split CIFAR10 setting, we firstly reproduce Split MNIST results and then expand to Split CIFAR10 setting based on open-source codes. More details such as hyperparameters and network structures of our model are described in Appendix A.

## 6.1 CONDITIONAL VS UNCONDITIONAL GENERATION

The quality of generated images is crucial in continual learning with the generative replay. In the generative replay, a model learns a new task over both the generated previous task data and the real current task data. If generated images have bad quality, then a classifier and a generator of the model degrade over the previous task data. Also in learning the next new task, the degraded generator generates poorer images, and the classifier degrades repeatedly. Therefore generating images of good quality prevents the model from degrading in continual learning. Recently, this fact is also studied by Lesort et al. (2018b). Optimizing the VAE according to the conditional likelihood of data generates sharper and more realistic image samples (Sohn et al. (2015); van den Oord et al. (2016)). To double-check whether the conditional generation is effective, we generated a few MNIST samples conditionally or unconditionally (Fig. 2). Samples generated conditionally were shown to be more recognizable. Such images of good quality generated conditionally will eventually prevent the VAE from degrading in continual learning.

## 6.2 INCREMENTAL LEARNING WITH MNIST

In the Hsu et al. (2018), they categorized various incremental learning tasks as three general cases: incremental task, incremental domain, and incremental class learning. In this section, we address the incremental domain and class learning based on MNIST dataset. Since MNIST dataset is simple for a generative model to make highly realistic samples, we do not apply the domain translation trick to MNIST variant settings. The detailed explanation about each setting is described in Appendix D, and we recommend to read Appendix D for a deeper understanding of settings and reported results.

Table 2: Ablation experiment for effectiveness of domain translation on Split CIFAR10 setting.

| | 2 tasks w/o DT | 2 tasks w/ DT | 5 tasks w/o DT | 5 tasks w/ DT |
|---|---|---|---|---|
| DGR | **48.71** | 70.33 | 21.75 | 29.75 |
| RtF | 48.13 | 67.71 | 21.53 | 38.27 |
| OCDVAE | 48.48 | 67.84 | 21.64 | 33.15 |
| DiVA | 48.34 | **78.93** | **28.52** | **42.11** |

**Permuted MNIST** In this task, pixels of each image are shuffled with task-wise fixed permutations (Goodfellow et al. (2013); Kirkpatrick et al. (2017)). We applied five different permutations to original MNIST data to make 5 Permuted MNIST tasks. Most of the CL algorithms show high accuracy in this task. Our DiVA shows slightly low accuracy than others. This is because Permuted MNIST looks random noises and does not have spatial correlations between neighbor pixels. Thus, it is hard to generate such images with a CNN architecture exactly. For experience replay (ER) based methods such as VCL or GEM, we set memory size for ER to 200 per each class; This memory size is also used for the Split MNIST setting. This memory size corresponds to the additional memory requirement for DiVA, a decoder network, described at Table 4 in Appendix. While this task is addressed in most of the continual learning studies, Farquhar & Gal (2018) insightfully pointed out drawbacks of the Permuted MNIST task in that unrealistic large differences in each permuted digits lead to artificially lessened forgetting. Thus, we also report results for Split MNIST setting in the next subsection.

**Split MNIST** Split MNIST is more challenging than Permuted MNIST task in that added new tasks could have similar patterns with old tasks (more realistic differences). In this experiment, we consider the five subsets of the MNIST data: the first subset consists of {0, 1}, the second subset is composed of {2, 3}, and in the same manner for remaining subsets. Also, we set our network with a single-headed form of network Farquhar & Gal (2018), which makes predictions over all classes (digits 0 to 9) with one output head; a multi-headed network (Nguyen et al. (2018); Zenke et al. (2017); Lopez-Paz et al. (2017)) for Split MNIST comprises multiple output heads and uses a task id guiding to a specific head for its task, which is an easier setting than single-headed Split MNIST. We can see that WR only approaches such as EWC, SI, and IMM fail to prevent catastrophic forgetting for previous task data, see Figure 3. This is because they do not have any explicit terms to discriminate the old and similar new tasks in their loss function. Also considering minimization of cross entropy with softmax, values of newly added output nodes should be bigger than nodes for old tasks. Concurrently, weights connected to nodes for old tasks are trained to output small values. That is why WR only approaches fail in the Split MNIST setting without reference samples of the previous task. From Table 1, the VCL and GEM use previous real data stored in coreset memory. So they can maintain performance for previous tasks to some degree while generative replay approaches can effectively mitigate the catastrophic forgetting problem in this setting. This means that if a generator can generate samples which are almost same with real images, GR-based approaches could be a strong way to mitigate catastrophic forgetting. Furthermore, our proposed model achieves higher accuracy than other GR-based methods. This is because our model conditionally generates previous samples, which makes more class-wise clear samples than other unconditional generative models.

## 6.3 INCREMENTAL LEARNING WITH CIFAR10

GR-based methods can outperform other CL approaches such as WR or ER in the challenging situation that an agent can observe each sequential task only once. The assumption of the advantage is that generated images should be highly similar to real images. However, there is a discrepancy between generated and real data distributions depending on the complexity of images. As a result, GR-based algorithms with a natural image dataset are suffering from catastrophic forgetting more than ER-based methods. In this section, we provide strong potential for GR-based approaches to be able to extend to natural image datasets.

**Split CIFAR10** This setting is the same as the Split MNIST task for CIFAR10 dataset. We divide CIFAR10 data into two or five subsets to make 2 or 5 class incremental tasks. To reduce the discrepancy of real and sample distributions, we used the r2s domain translation trick based on CycleGAN (Zhu et al. (2017)). The results are reported in Table 2. Without domain translation, all

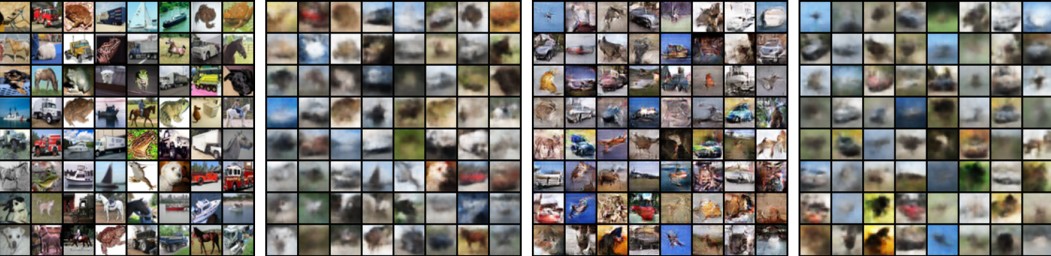

Figure 4: From left to right, real, r2s translation, s2r translation, and sample images generated by DiVA are displayed.

GR-based approaches suffer from serious catastrophic forgetting problem. However, by applying r2s translation, all algorithms show significant improvement on both 2 and 5 tasks Split CIFAR10 settings. Especially, DiVA achieved the highest accuracy among the state-of-the-art GR algorithms. Since all other GR algorithms have similar network capacities with DiVA, we can conclude that class-wise clear sample generation is important for incremental learning. Example images of domain translation are illustrated in Figure 4. We observed that the domain translator successfully translates the domain of real images into the domain of sample images generated by DiVA. This simple but powerful domain translation trick can be applied to any other GR-based approaches. Also, considering the improved quality of s2r translation, we think this technique can be further studied for a VAE-GAN hybrid model to take advantages of both models: diversity and high quality.

**Comparison of Domain Translation Tricks** Here, we compare possible domain translations described in Sec.5. In addition to our proposed domain translation tricks, Razavi et al. (2019) reported that applying reconstruction to test set images can significantly improve the CAS score (Ravuri & Vinyals (2019)). Since CAS score measures accuracy for real test set images of a model that trained with generated samples from a generative model, the reconstruction process can be viewed as a kind of domain translation. The results are at Table 3. The experiments were conducted in two settings: Split CIFAR10 divided

Table 3: Comparison of domain translation tricks with DiVA. Results are averaged after each setting is all done.

|  | 2 tasks | 5 tasks |
|---|---|---|
| Reconstruction | 75.89 | 38.31 |
| CycleGAN s2r | 72.41 | 19.54 |
| CycleGAN r2s | **78.93** | **42.11** |

into 2 and 5 subsets. Firstly, s2r translation showed lower accuracy than r2s translation. This is because that translating sample images to real style is still challenging even with an additional domain translator. While, r2s translation could be achieved more easily by, for example, blurring. Also, r2s translation showed higher accuracy than the domain translation using reconstruction. This is because distributions of reconstructed real images and randomly generated samples also have a discrepancy. In contrast, we directly model the translation between generated samples and real images. Furthermore, domain translation with reconstruction can only be applied to autoencoder-based architectures, while proposed methods can be applied to any kind of GR models, including GANs.

## 7  CONCLUSION

In this paper, we addressed how to mitigate the catastrophic forgetting problem. To solve the forgetting problem, we considered a generative replay based approach which generates previous task data and used them as a reference for previous tasks. We proposed a new type of conditional generative model based on the VAE architecture, which can be viewed as a classifier integrated generative model. Thus, we can conduct both the classification and the class conditional generation with one model. Also, we apply a simple domain translation trick based on CycleGAN to overcome the weakness of a generative replay based approach. As a result, our DiVA outperforms other recent continual learning algorithms in Split MNIST and CIFAR10 settings. We are sure that our DiVA has the potential to achieve higher accuracy than current reported results if a powerful generative model based on VAE is developed. Besides, we think that proposed domain translation trick can help other future continual learning researches which use the generative replay.

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

## A  APPENDIX

### A. IMPLEMENTATION DETAILS

Here, we describe the detailed network structure used in our experiments. For the EWC, we stacked two hidden layers, where each layer consists of 800 hidden units with ReLU activation. Also, we applied the dropout to the input layer and hidden layers with 0.2 and 0.5, respectively. We trained this EWC with batch size 256 and 50 epochs for each task. For fine-tuning with respect to $\lambda$, we conducted multiple runs with $\lambda = 10^3, 10^4, 10^5, 10^6, 10^7$ and selected the best value $\lambda = 10^6$. For the VCL, we chose the same structure as Nguyen et al. Nguyen et al. (2018). For the DGR and VGR, we carefully referred to the reported accuracy from their original papers. Other results that we reported in Table 1 are from Hu et al. (2019); Hsu et al. (2018); van de Ven & Tolias (2018). In our method, we found out that adding Gaussian noise to input images is useful to improve our algorithm for Split MNIST task. Since our loss function includes reconstruction error, there are small differences in pixel values between real images and generated images. So, this error can pass to next tasks and will be accumulated. Thus, the added Gaussian noise plays a positive role in training our model to become robust to the error. Adding noise technique is not necessary to reduce catastrophic forgetting for the prior-focused approaches because they do not include such generation process. GAN-based models such as the DGR and the VGR do not need to add noise too. This is because their loss function to generate an image is just the classification error of a discriminator, not the pixel-wise mean-squared

Table 4: Implementation details of the DiVA for Permuted MINST and Split MNIST settings

| Network | Operation | Kernel or Input dims | Stride/Padding or Output dims | BatchNorm | Activation |
|---|---|---|---|---|---|
| Encoder | Convolution | $3 \times 3 \times 1 \times 32$ | 2/1 | √ | ReLU |
| | Convolution | $3 \times 3 \times 32 \times 64$ | 2/1 | √ | ReLU |
| | Convolution | $3 \times 3 \times 64 \times 128$ | 2/1 | √ | ReLU |
| | Convolution | $3 \times 3 \times 128 \times 256$ | 2/1 | √ | ReLU |
| | Fully-connected | $2 \times 2 \times 256$ | 256 | √ | ReLU |
| | Fully-connected | 256 | 128 | - | ReLU |
| | Fully-connected | 256 | 128 | - | Softplus |
| Decoder | Fully-connected | 128 | $2 \times 2 \times 128$ | √ | ReLU |
| | Convolution | $3 \times 3 \times 128 \times 128$ | 2/1 | √ | ReLU |
| | Convolution | $3 \times 3 \times 128 \times 64$ | 2/1 | √ | ReLU |
| | Convolution | $3 \times 3 \times 64 \times 32$ | 2/1 | √ | ReLU |
| | Convolution | $3 \times 3 \times 32 \times 1$ | 2/1 | - | Sigmoid |
| Prior Net | Fully-connected | 10 | 128 | - | ReLU |
| | Fully-connected | 10 | 128 | - | Softplus |
| Classifier | Fully-connected | 128 | 10 | - | Softmax |
| Hyper-parameters | Optimizer | Adam(lr=0.001, betas=(0.9, 0.999), eps=1e-08) | | | |
| | Batch size | 64 | | | |
| | Softplus | beta=1, threshold=20 | | | |
| | $\lambda$ for DiVA | Permuted: 5e+3, Split: 2e+2 | | | |
| | Gaussian noise | mu=0, std=0.3 | | | |
| | Epochs per task | Permuted: {100, 100, 200, 200, 200} | | | |
| | | Split: {100, 100, 200, 200, 200} | | | |

error or binary cross entropy. Also, when we conducted some experiments with this adding noise technique for the EWC and the VCL, there was no noticeable increase in final accuracy for 5 tasks incremental learning. The detailed structures of the DiVA for Permuted and Split MNIST settings are described in Table 4, and learning curves for Permuted and Split MNIST tasks are illustrated in Figure 3. Also, for Split CIFAR10 setting, we implemented our encoder network for DiVA based on https://github.com/xternalz/WideResNet-pytorch. Depth, widen_factor, and dropRate are 28, 4, and 0.5, respectively. Also, we set the latent dimension for mu and variance of DiVA as 1024 and stacked one fully connected layer for both prior and classification network. We did not apply random noise to the Split CIFAR10 setting. The decoder network is doubled input and output channels than described in Table 4. For experimental results, we used 60,000 images for training and 10,000 images for testing in MNIST variant settings. Also, we used 50,000 images for training and 10,000 images for testing in CIFAR10 variant settings.

## B. KL-DIVERGENCE BETWEEN TWO GAUSSIAN DISTRIBUTIONS

Since we train both prior and posterior distributions for latent variables, we describe the detailed formulation of KL-divergence between two arbitrary Gaussian distributions here:

$$
\begin{aligned}
KL[q(x)||p(x)] &= \int p(x)\log p(x)\,dx - \int p(x)\log q(x)\,dx \\
&= \frac{1}{2}\log(2\pi\sigma_2^2) + \frac{\sigma_1^2 + (\mu_1 - \mu_2)^2}{2\sigma_2^2} - \frac{1}{2}(1 + \log 2\pi\sigma_1^2) \\
&= \log\frac{\sigma_2}{\sigma_1} + \frac{\sigma_1^2 + (\mu_1 - \mu_2)^2}{2\sigma_2^2} - \frac{1}{2}.
\end{aligned}
\tag{8}
$$

## C. DOMAIN TRANSLATION

CycleGAN Zhu et al. (2017) is an image-to-image translation algorithm. It translates an image from a source domain $\mathcal{X}$ to a target domain $\mathcal{Y}$ without paired examples (e.g., a photo of New York City and a painting by Gogh). They proposed a cyclic translation technique using a GAN architecture to keep the consistency of shape and convert the style of an image. The loss function of CycleGAN consists of adversarial losses $\mathcal{L}_{GAN}$ and a cycle consistency loss $\mathcal{L}_{cyc}$ as follows:

$$
\begin{aligned}
\mathcal{L}(G, F, D_{\mathcal{X}}, D_{\mathcal{Y}}) &= \mathcal{L}_{GAN}(G, D_{\mathcal{Y}}, \mathcal{X}, \mathcal{Y}) + \mathcal{L}_{GAN}(G, D_{\mathcal{X}}, \mathcal{Y}, \mathcal{X}) + \lambda\mathcal{L}_{cyc}(G, F), \\
where, \mathcal{L}_{GAN}(G, D_{\mathcal{Y}}, \mathcal{X}, \mathcal{Y}) &= \mathbb{E}_{y\sim p_{data}(y)}\left[\log D_{\mathcal{Y}}(y)\right] + \mathbb{E}_{x\sim p_{data}(x)}\left[\log(1 - D_{\mathcal{Y}}(G(x)))\right], \\
\mathcal{L}_{GAN}(F, D_{\mathcal{X}}, \mathcal{Y}, \mathcal{X}) &= \mathbb{E}_{x\sim p_{data}(x)}\left[\log D_{\mathcal{X}}(x)\right] + \mathbb{E}_{y\sim p_{data}(y)}\left[\log(1 - D_{\mathcal{X}}(G(Y)))\right], \\
\mathcal{L}_{cyc}(G, F) &= \mathbb{E}_{x\sim p_{data}(x)}\left[\|F(G(x)) - x\|_1\right] + \mathbb{E}_{y\sim p_{data}(y)}\left[\|G(F(y)) - y\|_1\right],
\end{aligned}
\tag{9}
$$

where, $G$ and $F$ mean the mapping functions $G : \mathcal{X} \to \mathcal{Y}$ and $F : \mathcal{Y} \to \mathcal{X}$. Also, $\lambda$ controls the relative importance of the two objectives. As a result, CycleGAN aims to solve:

$$
G^*, F^* = \arg\min_{G,F}\max_{D_x, D_y}\mathcal{L}(G, F, D_{\mathcal{X}}, D_{\mathcal{Y}}).
\tag{10}
$$

In our paper, the source domain $\mathcal{X}$ and the target domain $\mathcal{Y}$ mean real images and generated images, respectively. We used codes for CycleGAN from https://github.com/aitorzip/PyTorch-CycleGAN and followed default hyperparameter settings. Between each task, we trained CycleGAN for 60 epochs.

## D. DETAILED EXPLANATIONS OF INCREMENTAL LEARNING SETTINGS

There are two commonly used operations from datasets to evaluate continual learning algorithms: pixel permutation and class splitting. The typically used dataset is MNIST LeCun et al. (1998), which is composed of $0 \sim 9$ hand-written digit classes. The Permuted MNIST setting includes ten digit classes, where each class consists of different pixel permutations of each digit class in MNIST dataset.

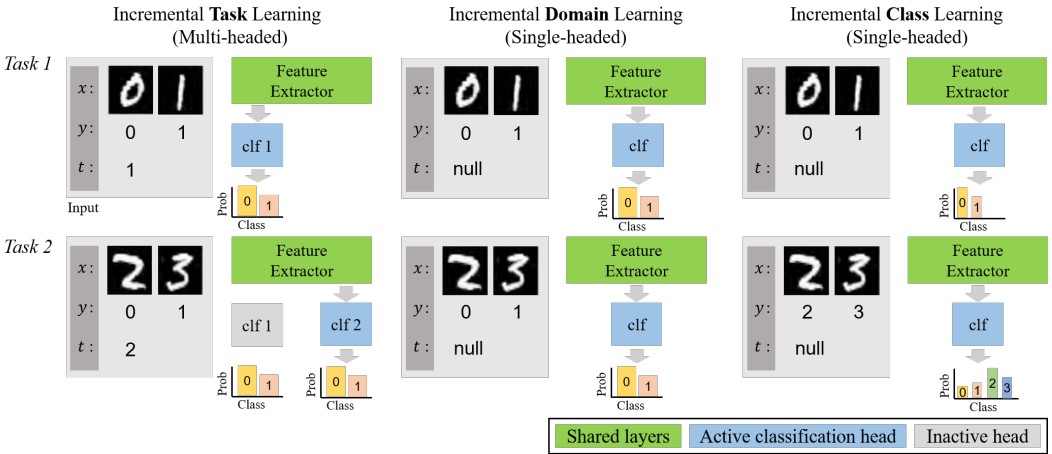

Figure 5: Three general incremental learning settings. This is based on Hsu et al. (2018); van de Ven & Tolias (2018)

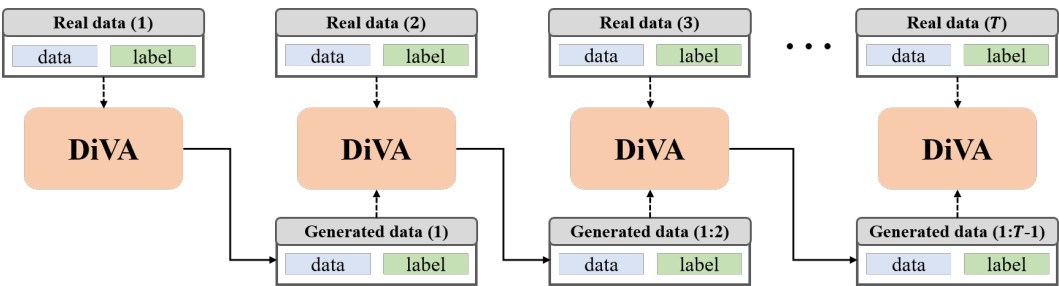

Figure 6: The process for incremental learning with the DiVA. At first, the model learns real data. Then, it generates all previous task data to learn a new task jointly. The number in () means each task.

This setting is widely used as incremental domain learning Kirkpatrick et al. (2017); Lee et al. (2017); Lopez-Paz et al. (2017); Nguyen et al. (2018); Shin et al. (2017); Zenke et al. (2017); Hu et al. (2019), despite some negative concerns that unrealistic substantial differences in each permuted digits lead to artificially lessened forgetting and, thus, making this setting less challenging Farquhar & Gal (2018). Next, the Split MNIST setting was initially introduced in a multi-headed from where the ten digits are split into five two-class classification tasks (the model has five out heads for each task) Nguyen et al. (2018); Zenke et al. (2017); Lopez-Paz et al. (2017). At the multi-headed architecture, each task has its own task id and independent classification heads. So, when training and testing a model, the task ids are given. This multi-headed split MNIST setting is also viewed as a relatively easy setting due to the task id. Farquhar and Gal Farquhar & Gal (2018) propose a single-headed variant which does not require task id, where a model should predict all classes that have seen. This single-headed Split MNIST is known as incremental class learning. Both our Split MNIST and CIFAR settings consider this single-headed Split setting which is more challenging than the multi-headed Split setting. A conceptual explanation of each setting is illustrated in Figure 5.

E. CONTINUAL LEARNING PROCESS OF DIVA

Here, we explain the whole process of DiVA for the incremental class learning task. At first, we train DiVA for task 1 with real images and labels. When task 2 comes, DiVA generates pseudo data and labels for task 1. Then, we train DiVA with both task 1 and task 2 jointly. When task 3 comes, DiVA

generates data and labels for task 1 and 2. Then, we train DiVA with pseudo data and labels for task 1 and 2, and real data and labels for new task 3. The conceptual explanation is illustrated at Figure 6.

