# OpenReview forum: "Discriminative Variational Autoencoder for Continual Learning with Generative Replay"
_ICLR.cc/2020/Conference — Reject_

### Official Review · AnonReviewer2 · 2019-10-16
**Official Blind Review #2**

**Rating:** 1

**Review:**

The paper devises a pipeline that aims to address catastrophic forgetting in continual learning (CL) by the well-known generative replay (GR) technique. The key ingredient of the pipeline is a modern variational auto-encoder (VAE) that is trained with class labels with respect to a mutual information maximization criterion.

The paper does not follow a smooth story line, where an open research question is presented and a solution to this problem is developed in steps. The flowchart in Fig 1 is rather a system design consisting of many components, the functionality of which is not clearly described and existence of which is not justified. This complex flowchart does not even describe the complete task. It is in the end plugged into a continual learning algorithm which also performs domain transformation. All of these pieces are very well-known methods (e.g. VAEs, conditional VAEs, CL, catastrophic forgetting, domain transformation) in the literature and this paper puts them together in a straightforward way. Hence, I kindly do not think the outcome is truly a research result. It is more system engineering than science.

The next submission of the paper could choose one or few of these pieces as target research problems and develop a thoroughly analyzed novel technical solution for them. If this solution can be proven to improve a valuable metric (e.g. accuracy, interpretability, theoretical understanding, or computational efficiency) of a setup, it is then worthwhile being published.

Minor: The abstract could be improved by providing more clear pointers to the presented novelty.

**Experience Assessment:**

I have published one or two papers in this area.

**Review Assessment: Checking Correctness Of Derivations And Theory:**

I assessed the sensibility of the derivations and theory.

**Review Assessment: Checking Correctness Of Experiments:**

I assessed the sensibility of the experiments.

**Review Assessment: Thoroughness In Paper Reading:**

I made a quick assessment of this paper.

---

> ### Author Response · Authors · 2019-11-10
> **Thank you for your valuable comments.**
>
> We appreciate your constructive feedback. Specifically, your comments about our motivation and development of our idea greatly help us to improve the quality of our paper.
>
> If we correctly understand reviewer 2’s concerns, the concerns can be divided into two folds:
>
> 1. Our suggestion to mitigate the catastrophic forgetting looks a naive combination of well-known concepts. Thus, it is more system engineering than science.
> 2. Each component described in Figure 1 is not explained enough. Also, there is no description of the complete task.
>
> [Response for 1]
> As we explained at the common response, we started our research from clear open questions. Our first open question was that why other GR-based algorithms [1, 2] assume unit Gaussian priors even though they integrate classification loss into their VAE formulation. Since they do not consider the conflict between the unit Gaussian prior and discriminative loss for the latent variable z, their models generate ambiguous samples that negatively affect the performance of incremental learning, which is discussed in section 4.1 in our paper. This leads us to a more theoretical formulation for classification-regularized VAE. By introducing class conditional priors induced by the mutual information maximization, DiVA yields class-wise discriminative one mode Gaussians for latent variable z. Naturally, DiVA can conduct both class prediction and class conditional sample generation with one integrated model.
>
> The second open question was that why GR-based algorithms suffer from serious catastrophic forgetting in natural image datasets, even though generated samples are not completely noisy. We assumed that this is due to the vulnerability of neural networks [3] triggered by different distributions of pixel values between real and generated images. Thus, we defined the two domains: real domain and sample domain. To narrowing the distribution gap, we needed a solution that satisfies two conditions (also described in section 5):
>
> 1. We should translate only the style (a global pattern of a specific domain) as keeping outline patterns of given images.
> 2. We should consider an unpaired domain translation between real and generated images because the generated images are sampled randomly.
>
> Fortunately, we were able to find an existing solution that satisfies the requirements: CycleGAN. Any other domain translators that satisfy the conditions can be used or newly studied. With the solution, we could make a breakthrough for GR-based methods. To the best of our knowledge, this is the first successful approach for a GR-based algorithm to start to resist the catastrophic forgetting problem with a natural image dataset.
>
> [Response for 2]
> Figure 1 is a conceptual description of our proposed model, DiVA. Each component is explained in section 4, below Equation 2, and justified in section 4.1. Also, for an easy understanding of the whole CL process with DiVA, we added another figure in Appendix E.
>
> [References]
> [1] van de Ven, Gido M., and Andreas S. Tolias. "Generative replay with feedback connections as a general strategy for continual learning." arXiv preprint arXiv:1809.10635 (2018).
>
> [2] Mundt, Martin, et al. "Unified Probabilistic Deep Continual Learning through Generative Replay and Open Set Recognition." arXiv preprint arXiv:1905.12019 (2019).
>
> [3] Su, Jiawei, Danilo Vasconcellos Vargas, and Kouichi Sakurai. "One pixel attack for fooling deep neural networks." IEEE Transactions on Evolutionary Computation (2019).

---

### Official Review · AnonReviewer3 · 2019-10-22
**Official Blind Review #3**

**Rating:** 1

**Review:**

-- This paper seeks to combine several ideas together to propose an approach for image classification based continual learning tasks. In this effort, the paper combines previously published approaches from generative modeling with VAEs, mutual information regularization and domain adaptation.

I am a making a recommendation for reject for this paper with the main reason being that I believe the primary derivations for their method appear flawed.

--In the main section describing the approach (Section 4), the authors start with a claim that Equation 1 and 2 are equal; I don’t believe 1 and 2 are equal.

--In Section 4.1, it appears that they are instead making a claim about Equation 2 being a bound for equation 1; but even this derivation appears to have a problem. The following is the concern:

--In the second line of Equation 5, the KL term appears to be measuring a distance between distributions on two different variables; z|c and c|z. If one were to interpret the second one as the unnormalized distribution on z defined via the likelihood for c given z; even this has an issue because then the expression for KL where we plug the unnormalized density in place of the normalized need not be positive which is something they need to derive their bound.

--Another issue is that the regularization lambda should apply to both the terms in the bound but in Equation (7) only appears selectively for one of the two terms.

It is also not clear how the loss function proposed differs from that of the CDVAE, etc.  If the novelty is in applying to continual learning and new datasets, it is not clear that this is sufficient.

Additional feedback for authors (not part of the main decision reasoning):

- What is dt in Algorithm 1 description?

Figure 1:
-typo “implmented”
-What’s the 3d plot supposed to represent?
Doesn't the classification loss have a dependency on the input condition?

--What does a "heavy classifier" imply concretely?

--“Redundant weights” seems like not a very strong constraint especially for a small cardinality label space (like 10, in the case of this paper).

--The notation for the proposed parameters theta, theta’, phi, phi’ are not consistent with the notation in the intro section, where phi was used for the encoder and theta for the decoder. In later sections they use theta and theta’ for encoder/decoder resp.

-- “When the encoder and decoder networks are sufficiently complex, it is enough to implement each the prior and classification network as one fully-connected layer” → what do the authors mean “when … networks are sufficiently complex” or do they actually mean when the “when the problem is simple enough”?


**Experience Assessment:**

I do not know much about this area.

**Review Assessment: Checking Correctness Of Derivations And Theory:**

I assessed the sensibility of the derivations and theory.

**Review Assessment: Checking Correctness Of Experiments:**

I did not assess the experiments.

**Review Assessment: Thoroughness In Paper Reading:**

I made a quick assessment of this paper.

---

> ### Author Response · Authors · 2019-11-10
> **Thank you for your valuable comments.**
>
> We appreciate your constructive feedback. Specifically, your comments about our derivations greatly help us to improve the quality of our paper. We hope you to also consider our notable experimental results as well.
>
> (Bounds of KL divergence) Thank you for this good comment. We claimed that the Equation 1 can be maximized indirectly by maximizing Equation 2 which is a lower bound of Equation 1. If we understand your primary concern correctly, the concern comes from the bound of KL divergence in Equation 5. To prove correctness of our formulation, we can rewrite the pointed term in Equation 5 by using simple bayes rule as follows:
>
> $$\displaystyle\sum_{\mathrm{z}}\hat{q}\mathrm{(z|c)}\ \mathrm{log}\ \frac{\hat{q}\mathrm{(z|c)}}{\hat{p}\mathrm{(c|z)}} = \displaystyle\sum_{\mathrm{z}}\hat{q}\mathrm{(z|c)}\bigg(\mathrm{log}\ \frac{\hat{q}\mathrm{(z|c)}}{\hat{p}\mathrm{(z|c)}} + \mathrm{log}\ \frac{\hat{p}\mathrm{(z)}}{\hat{p}\mathrm{(c)}} \bigg)$$
>
> Because the $\hat{p}\mathrm{(c)}$ is constant, and $\hat{p}\mathrm{(z)}$ is not included in our optimization, we just optimize $\displaystyle\sum_{\mathrm{z}}\hat{q}\mathrm{(z|c)}\mathrm{log}[\hat{q}\mathrm{(z|c)} / \hat{p}\mathrm{(z|c)}]$. Since the $\hat{q}\mathrm{(z|c)}$ and $\hat{p}\mathrm{(z|c)}$ are both normalized distributions, the $D_{KL}[\hat{q}\mathrm{(z|c)} || \hat{p}\mathrm{(z|c)}]$ is always positive. Then, we can conclude that Equation 2 becomes the lower bound for Equation 1.
>
> (lambda)  Actually, Equation 7 consists of three terms. Since only the third term is proposed additional regularization, we applied weighting parameter lambda to the third term only.
>
> (Difference with CDVAE) To clarify the difference our DiVA with CDVAE, we write derivations for both models here.
>
> CDVAE: $\mathbb{E}_{q_{\theta}\mathrm{(z|x)}}[\mathrm{log}\ p_{\theta '}(\mathrm{x|z)}] - D_{KL}[q_{\theta}\mathrm{(z|x)} || p\mathrm{(z)]} + \lambda \mathbb{E}_{q_{\theta}\mathrm{(z|x)}}[\mathrm{log}\hat{p}_{\phi '}\mathrm{(c|z)}]$
>
> DiVA: $\mathbb{E}_{q_{\theta}\mathrm{(z|x)}}[\mathrm{log}\ p_{\theta '}(\mathrm{x|z)}] - D_{KL}[q_{\theta}\mathrm{(z|x)} || \hat{q}_{\phi}\mathrm{(z|c)]} + \lambda \mathbb{E}_{q_{\theta}\mathrm{(z|x)}}[\mathrm{log}\ \hat{p}_{\phi '}\mathrm{(c|z)}]$
>
> As we discussed in section 4.1, below the table for Algorithm 1, the key difference is that we consider class-conditional Gaussian distributions as priors for variational posteriors. Since CDVAE assumes the prior as unit Gaussian for all classes and optimizes classification loss simultaneously with the KL divergence, the latent space does not follow the prior exactly. As a result, CDVAE sometimes generates ambiguous samples (Figure 2 (c)). Interestingly, RtF [1] also does not consider the class-conditional priors even though they consider a classifier integrated VAE similar to CDVAE. In contrast, we assume class-wise specific Gaussian for each class. As a result, we can stably generate more realistic samples than CDVAE.
>
> [Additional feedback]
> (dt in Algorithm 1) dt means the domain translation explained at section 5.
>
> (Figure 1)
> - We corrected the typo.
> - The 3d plot conceptually represents class-specific one mode Gaussians.
> - The classification loss has implicit dependency with input conditions by minimizing the KL divergence in Equation 2.
>
> (heavy classifier) A classifier such as resnet. We used this term to distinguish the additional classifier from our integrated encoder that has discriminative power.
>
> (Redundant weights) If we extend to a more complex dataset such as ImageNet, it will become highly redundant. Furthermore, if we consider fully-convolutional architecture (without fully-connected layers), redundancy becomes a serious problem. For example, a feature map that has shape of [W x H x dim] becomes [W x H x (dim + the number of classes)]. In contrast, using discriminative conditional distributions can keep the dimension of the feature map as [W x H x dim] regardless of the number of classes.
>
> (Notations) Thank you for commenting this. We corrected the notations of section 3 to match with later sections.
>
> (Complexity of encoder) We intended that the encoder network can have enough both discriminative and generative power with a powerful architecture such as a deep residual network.
>
> [References]
> [1] van de Ven, Gido M., and Andreas S. Tolias. "Generative replay with feedback connections as a general strategy for continual learning." arXiv preprint arXiv:1809.10635 (2018).

---

### Official Review · AnonReviewer1 · 2019-10-23
**Official Blind Review #1**

**Rating:** 3

**Review:**

In this paper, the authors focus on alleviating the catastrophic forgetting problem in continual learning.  The authors propose a discriminative variational autoencoder (DiVA) to solve this problem under the generative replay framework. DiVA modifies the objective function of VAE by introducing an additional term that maximizes the mutual information between the latent variables and the class labels.

The authors do not thoroughly explain the motivation of this paper. The authors do not explicitly define continual learning, incremental learning, and catastrophic forgetting problem. It is also not clear to me why these problems are important.

The idea that introduces labels in VAE is not novel. For example, Narayanaswamy et al. [1] also propose to utilize labels to VAE. I do not understand why making use of labels is important for solving the catastrophic forgetting problem and how the labels are useful in the generative replay process. It is also not clear to me how domain translation is relevant to continual learning.

In terms of modeling, since the input into the prior network has finite possible discrete values, we do not need a fully connected network to generate $\hat{\mu}_c$ and $\hat{\sigma}_c$. Instead, we can directly optimize $\hat{\mu}_c$ and $\hat{\sigma}_c$ for each $c$ as parameters.

The paper provides some good experimental results. But the problem settings are not clear to me. I do not understand how the model is trained to solve multiple tasks. Do the same model is trained for multiple tasks? Is each of the tasks trained sequentially or simultaneously? It is also not clear to me why CIFAR datasets involve two domains and how these domains are relevant in each of the tasks.

In summary, since DiVA gives a good experimental performance, the proposed method might be promising. However, it looks to me that the authors need to better explain the motivation of DiVA, the differences of DiVA from existing supervised VAE, and the experimental settings, before the acceptance of this paper.

References
[1]Narayanaswamy, Siddharth, T. Brooks Paige, Jan-Willem Van de Meent, Alban Desmaison, Noah Goodman, Pushmeet Kohli, Frank Wood, and Philip Torr. "Learning disentangled representations with semi-supervised deep generative models." In Advances in Neural Information Processing Systems, pp. 5925-5935. 2017.

**Experience Assessment:**

I do not know much about this area.

**Review Assessment: Checking Correctness Of Derivations And Theory:**

I carefully checked the derivations and theory.

**Review Assessment: Checking Correctness Of Experiments:**

I assessed the sensibility of the experiments.

**Review Assessment: Thoroughness In Paper Reading:**

I read the paper at least twice and used my best judgement in assessing the paper.

---

> ### Author Response · Authors · 2019-11-10
> **Thank you for your valuable comments.**
>
> We appreciate your constructive feedback. Specifically, your comments about the motivation and problem definitions greatly help us to improve the quality of our paper.
>
> (Importance and motivation) To step forward to artificial general intelligence, we should further consider making an agent that can learn and remember many tasks incrementally [1]. However, this is particularly challenging in real-world settings: the agent may observe different tasks sequentially, and an individual task may not recur for a long time. In this settings, a learned model might overfit to the most recently seen data, forgetting the rest, a phenomenon referred to as catastrophic forgetting, which is a core issue CL systems aim to address [2]. Recently, GR-based methods, inspired by the generative nature of the hippocampus as a short-term memory system in the primate brain [3], have been widely studied to address the catastrophic forgetting problem. In terms of GR, we are trying to address the two open questions mentioned above.
>
> (Use of labels and novelty) In GR-based approaches, the quality of generated samples is crucial to keep the performance of previous tasks. If we use labels, we can construct a conditional generative model. Generally, conditioning on a generative model yields higher quality samples than unconditional one and makes it possible to generate class-balanced samples [4]; the importance of conditional generation is also described in section 6.1 in our paper. In this paper, we showed that discriminative regularization could make VAE possible to conduct both class conditional generation and classification with one integrated model. Thus, we do not need to train an additional classifier, e.g., deep CNN, which is necessary for other works, including Narayanaswamy et al. There is also classifier integrated VAE such as [6]. The difference with [6] is the use of class-conditional priors; more details are explained at the response (Difference with CDVAE) for reviewer 3 and section 4.1 in our paper.
>
> (Domain translation) Even though the conditional generation improves the quality of the generated samples, there is still a big difference between real and generated images. Because a deep neural network is vulnerable to even single-pixel perturbation [5], the difference can seriously affect the classification performance of GR-based algorithms. Thus, we suggested applying the domain translation to address this issue. By narrowing distribution discrepancy between real and generated images using the domain translation technique, we were able to alleviate the catastrophic forgetting problem successfully (Table 2).
>
> (Modeling) Good point. Since we consider finite discrete conditions, we can directly optimize $\mathrm{\mu_c}$ and $\mathrm{\sigma_c}$ for each c as parameters without the prior network. However, introducing a prior network makes our model become a more general framework that can address continuous-valued conditions. Also, in our paper, we set the prior network as a single fully-connected layer for easy handling of conditions and simple implementation. Otherwise, we should keep an additional mapping table between class conditions and its $\mathrm{\mu_c}$ and $\mathrm{\sigma_c}$.
>
> (Experimental settings)  Generally, CL systems assume that each task comes sequentially, and an agent can not directly access previous experience [2]. We exactly follow the assumption. Also, we train DiVA sequentially for each task with one same model. To clarify our training process, we provide a brief summarization. Firstly, we train DiVA with task 1 that consists of real images and labels. Then, when new task 2 is coming, DiVA generates images and its labels of task 1 and learns both task 2 and the generated task 1 simultaneously. We added an additional figure in Figure 6 in Appendix E, for helping conceptual understanding.
>
> (Domains of CIFAR dataset) Since current generative models are not perfect for generating complex natural images, there is always a discrepancy between generated images and real images. Thus, we can define two domains: real image domain (realistic) and generated image domain (blurry). We used the domain translation for narrowing the gap.

---

> > ### Author Response · Authors · 2019-11-10
> > **[References]**
> >
> >
> > [1] Legg, Shane, and Marcus Hutter. "Universal intelligence: A definition of machine intelligence." Minds and machines 17.4 (2007): 391-444.
> >
> > [2] https://sites.google.com/view/continual2018
> >
> > [3] Shin, Hanul, et al. "Continual learning with deep generative replay." Advances in Neural Information Processing Systems. 2017.
> >
> > [4] Lesort, Timothée, et al. "Marginal Replay vs Conditional Replay for Continual Learning." International Conference on Artificial Neural Networks. Springer, Cham, 2019.
> >
> > [5] Su, Jiawei, Danilo Vasconcellos Vargas, and Kouichi Sakurai. "One pixel attack for fooling deep neural networks." IEEE Transactions on Evolutionary Computation (2019).
> >
> > [6] Mundt, Martin, et al. "Unified Probabilistic Deep Continual Learning through Generative Replay and Open Set Recognition." arXiv preprint arXiv:1905.12019 (2019).

---

> > ### Comment · AnonReviewer1 · 2019-11-15
> > **Thank you for your response**
> >
> > Thank you for your response. After checking the literature more carefully, I realized that there have been generative replay models based on conditional GAN [1]. It avoids the problem of the proposed model that it needs to first generate images based on conditional VAE and then convert them into better quality images via cycle-GAN. Conditional GAN can directly generate high-quality images in one step. I do not suggest acceptance of this paper unless experimental results are provided showing that the proposed model outperforms the conditional-GAN-based model.
> >
> > References
> > [1] Wu, Chenshen, et al. "Memory replay GANs: Learning to generate new categories without forgetting." Advances In Neural Information Processing Systems. 2018.

---

> > > ### Author Response · Authors · 2019-11-15
> > > **Thank you for your feedback**
> > >
> > > Thank you for your feedback. Since, the rebuttal deadline is almost end, we are not sure to report the additional comparison that you raise. Nevertheless, we are now going to start the CL experiment with the [1] based on CIFAR 10 dataset.  However, we hope to say that the DGR in our paper is also based on the WGAN-GP [2] which can generate high-quality images. The experimental result on Table 2 shows that the WGAN-GP based GR algorithm also suffers from severe catastrophic forgetting on CIFAR 10 dataset.
> > >
> > >
> > > [References]
> > > [1]Wu, Chenshen, et al. "Memory replay GANs: Learning to generate new categories without forgetting." Advances In Neural Information Processing Systems. 2018.
> > >
> > > [2] Gulrajani, I., Ahmed, F., Arjovsky, M., Dumoulin, V., & Courville, A. C. (2017). Improved training of wasserstein gans. In Advances in neural information processing systems (pp. 5767-5777).

---

### Author Response · Authors · 2019-11-10
**[Response for common concerns]**

We greatly appreciate all reviewers for valuable concerns and constructive feedback to improve clarity and quality of our paper. We will firstly respond common concerns about motivations, then address main points that each reviewer raised.

In this paper, we primarily address two issues: “what are current GR-based methods missing?” and “how can we extend GR-based CL approaches to a more complex dataset such as CIFAR10?”. We suggested two solutions to answer the open questions: a new type of conditional VAE that can also predict class labels (DiVA) and applying a domain translation (DT) trick. By applying DT, we significantly improved the continual learning performance of the current state of the art GR-based algorithms (Table 2). Furthermore, DiVA achieved the highest CL accuracy among the GR-based algorithms. We believe that this could be an important step that will trigger other GR-based researches trying to address more complex natural image datasets.

---

### Decision · Program_Chairs · 2019-12-19

**Decision:**

Reject

**Comment:**

The paper presents a method for continual learning with a variant of VAE. The proposed approach is reasonable but technical contribution is quite incremental. The experimental results are limited to comparisons among methods with generative replay, and experimental results on more complex datasets (e.g., CIFAR 100, CUB, ImageNet) are missing. Overall, the contribution of the work in the current form seems insufficient for acceptance at ICLR.